# Reliability and Validity of a Stress Scale in Public Employees from Murcia (Spain)

**DOI:** 10.3390/ijerph17238894

**Published:** 2020-11-30

**Authors:** María Teresa Rodríguez-González-Moro, Juana Inés Gallego-Gómez, José Miguel Rodríguez-González-Moro, María Consolación Campillo Cano, José Miguel Rivera-Caravaca, Agustín Javier Simonelli-Muñoz

**Affiliations:** 1Faculty of Nursing, Catholic University of Murcia (UCAM), 30107 Murcia, Spain; mtrodriguez@ucam.edu (M.T.R.-G.-M.); mcampillo2@ucam.edu (M.C.C.C.); 2Department of Pneumology, Hospital Universitario Príncipe de Asturias, Alcalá de Henares, 28805 Madrid, Spain; respirama@yahoo.es; 3Department of Cardiology, Hospital Clínico Universitario Virgen de la Arrixaca, Instituto Murciano de Investigación Biosanitaria (IMIB-Arrixaca), CIBERCV, 30120 Murcia, Spain; jmrivera429@gmail.com; 4Department of Nursing Science, Physiotherapy and Medicine, Faculty of Health Sciences, University of Almería, 04120 Almería, Spain; sma147@ual.es

**Keywords:** stress, work-related stress, government employees, human characteristics

## Abstract

Stress is common in all work environments. Technostress and the difficulty of separating the family arena from the work environment are some of the new and emerging risks faced by companies, employees and society in general. Most of the available instruments for measuring stress in workers have been focused on education professionals and healthcare workers. Therefore, it is necessary to validate simple and friendly-use tools to detect stress levels in public workers. The aim of this study was to determine the internal consistency of an adapted version of the Student Stress Inventory-Stress Manifestations (SSI-SM) for public employees and to determine if high-stress levels are related to personal and work-related factors. A cross-sectional and descriptive study was conducted from October 2016 to February 2019 including 468 Spanish public workers based in Murcia. An adapted version of the SSI-SM was administered and data on personal and work-related factors were collected. Results showed that all of the factors had Cronbach’s α over 0.700, and no items need to be deleted due to correlations with the factor exceeding 0.300. Factor 1, *“Self-concept”,* has a Cronbach’s α of 0.868, with values of 15.62 ± 4.99; factor 2, *“Sociability”,* Cronbach’s α: 0.853, with mean values of 13.33 ± 4.17; factor 3, *“Somatization”*, Cronbach’s α: 0.704, mean value of 5.35 ± 1.90 and: factor 4, *“Uncertainty”,* Cronbach’s α: 0.746, with a mean value of 8.19 ± 2.51. In conclusion, the internal consistency of the adapted SSI-SM for public employees with different work positions and shifts has been validated and determined. This study provides a useful tool for the early detection of stress in public employees and may be potentially useful for preventing the harmful consequences of stress.

## 1. Introduction

Stress is defined according to three notions described in the literature: stress as a stimulus, stress as a response, and stress as an event-reaction relationship [1]. Under the psychological perspective of stress as a response, based on Selye’s General Adaptation Syndrome [2], stress generates physiological, emotional, behavioral reactions and negative effects in organizations. It may cause even burnout, as a response to chronic stress. Thus, work-related stress is of increasing interest in our daily lives, given its potential consequences on both employee health and business results (employment leave, absenteeism and poor performance) [3]. Labour has evolved from being mainly a physical activity to having an increasing mental load on the individual [4] and thus, it produces physiological, physical and psychological effects in the short and long-terms, such as negative mood and anxiety [5,6]. Factors such as responsibility, information processing, job-related uncertainty and role ambiguity [7], as well as technostress [8], have contributed to the emergence of new risks to workers’ mental health.

Despite that stress may have a positive effect on workers, acting as a motivational factor that improves creativity, it also may surpass the individual’s capabilities, leading to negative consequences [2,9]. Hence, it might drive to excessive costs that are associated with these consequences, becoming a problem for workers, companies, labour risk prevention services and the healthcare system in general [10]. Moreover, it is associated with low productivity and worse job satisfaction [11,12].

On the other hand, certain individual behaviors may lead to an increased level of stress and may decrease adaptative coping strategies. For example, it is well known that taking work home, access to Information and Communication Technology (ICT) resources at home [13], and overeating dinners [14], may negatively affect stress management. In addition, it should be noted that “all jobs are potentially stressful, although the stress level varies considerably, depending on the individual and his/her coping mechanisms” [15].

For the above reasons, identifying potential stressors and measuring stress is central for the appropriate management of workers’ mental health. Nevertheless, the evaluation of stress in workers is difficult due to the high complexity of an objective quantification. To date, different instruments have been validated for measuring stress in workers, but most of these have focused on education professionals and healthcare workers [16,17,18]. Thus far, we have not found questionnaires, aimed at all types of workers that focus on stress as a response. For example, the perceived Stress Scale (PSS), which is one of the best known and most used tools in Spain, has disparate results. Thus, a study showed that the first factor was responsible for 25.9% of the total variance, and the second for 15.7% [19]. In a more recent study, it was observed that the first factor explained 60.5% of the variance, and in conjunction with the second, they explained 81.1% of the variance [20]. The study by Minura and Griffiths in a Japanese population demonstrated that the first factor was responsible for 23.8% of the variance, and the second for 18.8% [21]. Nevertheless, Gonzalez and Landero found in a two-dimensional organization, the first factor explained 32.6% of the total variance, whereas the second explained 15.4% [22]. Therefore, in the assessment of constructs for which there is no gold standard, it is necessary to corroborate the psychometric behavior in populations with different characteristics [23].

Thus, the Student Stress Inventory-Stress Manifestations (SSI-SM) is a simple instrument for screening manifestations of stress as a response and has recently been validated in university students [24]. Our hypothesis is that such a concise instrument would be useful to detect stress as a response in workers. The objective of this study is to determine the internal consistency and validity of the Student Stress Inventory-Stress Manifestations questionnaire that was adapted to public employees and to determine whether high levels of stress in Spanish public employees are related to personal and work-related factors.

## 2. Materials and Methods

This is a descriptive and cross-sectional study including public employees from different departments of the Autonomous Community of Murcia (Spain), i.e., with different work positions, from October 2016 to February 2019. All public workers attending face-to-face routine medical examinations by the Risk Prevention Service were suitable for the study. The only exclusion criterion was the current treatment with psychotropic drugs.

At inclusion, the following information on personal and work-related factors of the public employees was collected: age, sex, civil status, body mass index (BMI) and work position. According to work position, the employees were divided into four categories: manual positions, including workers from the tertiary sector (i.e., Services sector: security guards, ordinance workers, kitchen assistants, cooks, cleaning personnel, transport workers, carpenters, plumbers, mechanics, electricians, etc.); administrative positions (office, administrative, judicial-legal, planning and computer system workers); technical positions (workers associated with university studies, graduate degrees or graduates) and; management positions (service managers, directors and general managers). In addition, a brief questionnaire with the following questions was provided: “*I think about work at home, or have concerns regarding work before going to bed*”, “*I do some home-based telework*”, “*I use to take a copious or oversized dinner*”, “*I use ICTs resources at home*” and “*I stay in bed, even when I cannot fall asleep*”. These questions were scored from 0 to 4 in a Likert-type score, with 0 being “never” and 4 being “always”.

The medical staff of the Risk Prevention Service recorded all data and was in charge of passing the questionnaires. They were previously informed of the characteristics of the study, as well as the purpose for which the data recorded will be used. Several meetings were held with the staff, explaining and clarifying the data collection method to avoid bias in the selection of the sample and in the measurement of the variables.

### 2.1. The Student Stress Inventory-Stress Manifestations (SSI-SM)

The SSI-SM questionnaire was described by Fimian et al. to quantify stress levels [25]. It was translated into Spanish in 2011 by Espejo and collaborators and its psychometric properties were described [26]. Ortuño-Sierra et al. analysed the psychometric properties and the invariance in a large sample of adolescents, obtaining results that indicated that the SSI-SM scores presented adequate psychometric properties [27]. Cronbach’s alpha coefficient for the subscales ranged between 0.69 and 0.90 and it has been recently validated in Spanish university students [24]. This questionnaire included 22 items using a 5-point Likert-type score (from 1 = not at all, to 5 = completely) related to emotional (10 items), physiological (6 items) and behavioral areas (6 items). The higher perceived stress, the higher score on the scale.

### 2.2. Ethical Considerations

The study was approved by the Ethics Committee of the Catholic University of Murcia (UCAM) (code: CE111707) and was carried out in accordance with the ethical standards established in the Declaration of Helsinki. The study started upon receipt of the authorisation of the General Management of Civil Service and all workers had to sign an informed consent to participate. The data collected were recorded and processed anonymously.

### 2.3. Statistical Analysis

The software Ene 2.0 (GlaxoSmithKline, Brentford, UK) was used to calculate the sample size based on an estimation of 45% of perceived stress [28], with an accuracy of ±5%, an α error of 5%, and for an infinite population. A minimum sample of 263 public employees was necessary.

Categorical variables were expressed as frequencies and percentages. Continuous variables were described using the median and the interquartile range or the mean and standard deviation, if the distribution was normal, in accordance with the Kolmogorov-Smirnov test.

To compare the association between the variables, a Pearson’s chi-square test was used, along with a Student’s *t*-test and a Pearson’s correlation.

To measure the internal consistency and homogeneity of the SSI-SM questionnaire, a coefficient of 0.700 was considered the ideal value for Cronbach’s α. The individual analysis of each item was carried out using the Homogeneity Index, which evaluated them using Pearson’s correlation coefficient. Each item with a coefficient > 0.300 was considered to be useful for assessing the attribute, excluding the items that do not comply with this condition. To analyse the underlying conditions present in the test, multivariable factor analysis was used. Before this analysis, the suitability of the data was analysed using the Kaiser-Meyer-Olkin test. The contrast of the correlation matrix was verified using Bartlett’s test of sphericity. The factorial analysis was performed by exploring the main components of the correlation matrix for each questionnaire item, with orthogonal rotation using Varimax rotation with Kaiser normalisation. Only factors with values higher than 1 were extracted, since these are the ones explaining the higher degree of the total variability, using the criteria that the extracted components make up at least 60% of the variance explained by the correlation matrix. In order for the factorial weights to be consistent, a criterion was established that for an item to form a part of the extracted factor, its value must be equal to or greater than 0.40.

A value of *p* < 0.05 was considered statistically significant. Statistical analyses were performed using SPSS 21.0 for Windows (SPSS, Inc., Chicago, IL, USA).

## 3. Results

We included 468 public employees from the Región de Murcia (Spain). The mean age was 47.6 ± 7.3 years and 51.1% were male (mean BMI 26.1 ± 4.3 kg/m^2^). The majority (75.9%) of workers were married or lived with their partner. As for the productive sector of the workers, 59.4% were technical workers having a university education, 20.5% held administrative positions, 12.6% were managers and 7.5% had manual jobs.

During the homogeneity analysis, two items were excluded (14 and 22) from the original SSI-SM since their correlation coefficient with the overall corrected scale was lower than 0.300. Then, a Cronbach’s α value of 0.909 was obtained. The mean value of the SSI-SM was 33.92 ± 9.3 (95% CI, 33.0–34.7), with a minimum value of 20 and a maximum of 66 points.

In order to determine the validity of the construct in the factorial analysis of the adapted SSI-SM questionnaire, first, it was determined that the criteria necessary for its application were met, verifying the existence of an underlying structure made up of four factors, in accordance with the Kaiser rule, which collectively explained 61.3% of the variance. The factorial load of each item was satisfactory for inclusion in the model since their values were >0.400. Following the rotation, factor 1 included eight items related to the “*Self-concept*” of personality, factor 2 included eight items related to “*Sociability*”, factor 3 consisted of four items that measure aspects defining the “*Somatization*” subject, felt by the public employees as a result of stressful situations, and factor 4 contained five items, analysing the “*Uncertainty*” related to aspects of insecurity (Table 1).

The homogeneity of the factors resulting from the factorial analysis was analysed. They all had Cronbach’s α values exceeding 0.700, and in no case was it necessary to eliminate items due to a correlation with its factor of over 0.300. Factor 1 “*Self-concept*”, had a Cronbach’s α of 0.868, with mean values of 15.62 ± 4.99; factor 2 “*Sociability*”, a Cronbach’s α of 0.853, with mean values of 13.33 ± 4.17 points; factor 3 “*Somatization*”, a Cronbach’s α of 0.704, with a mean of 5.35 ± 1.90 points, and factor 4 “*Uncertainty*”, a Cronbach’s α of 0.746, with a mean value of 8.19 ± 2.51 points.

The mean score on the final version of the SSI-SM was 33.92 ± 9.32. Table 2 shows the associations between different behaviours in households with workers with the four factors and with the overall score of the stress questionnaire. Neither civil status nor BMI was associated with any of the factors. Upon analysing stress based on work category, it was found to be higher in technical workers having university studies for factor 3 “*Somatization*” (*p* = 0.009). There was also a significant difference in the same factor in terms of sex, with higher scores on “*Somatization*” in females (5.0 ± 1.6 vs. 5.6 ± 2.1, *p* = 0.001).

Regarding the personal and work-related factors, there were strong correlations between the items “*I think about work at home, or have concerns regarding work before going to bed*” (R = 0.374, *p <* 0.001), “*I do some home-based telework*” (R = 0.264, *p* < 0.001), “*I use to take a copious or oversized dinner*” (R = 0.200, *p* < 0.001) and “*I use ICTs resources at home* ” (R = 0.100, *p* = 0.031) with the score in the final version of the SSI-SM. In addition, comparing the median scores of the four factors of the SSI-SM with these four items, all were found to be statistically significant. On the other hand, there was no significant correlation between the item “*I stay in bed, even when I cannot fall asleep*” with neither, the overall score in the SSI-SM or any of the four analysed factors. In addition, we did not find relations between high levels of stress according to the adapted SSI-SM and sex, civil status, BMI or the work position.

## 4. Discussion

In 2011, Espejo et al. [26] translated to Spanish the SSI-SM scale developed by Fimian, Fastenau, Tashner and Cross [25], and they performed an analysis of its psychometric properties. This brief scale has been validated in adolescent and adult age university students [24], revealing very acceptable results. In the present study, we proposed validation of this simple tool in a sample of public employees. The SSI-SM measures stress from three areas: emotional, physiological and behavioural, which may help in detecting stress in public employees holding different work positions. After excluding two items from the original SSI-SM in the analysis of homogeneity, we found that it was useful for identifying stressors and stress symptoms in public workers. All of the factors presented an appropriate Cronbach’s α value, and in all of the items, there was a correlation with its factor that exceeded 0.300. 

### 4.1. Other Tools to Assess Stress

There are different tools to measure the manifestations of stress, however, all have a specific focus on different professions. The Teacher Stress Inventory (TSI) assesses the sources and manifestations of stress in teachers and has been validated and translated into several languages. Thus, in Pakistan, the global reliability coefficient was 0.85, and for the subscales, it ranged between 0.63 and 0.80 [29]. In Greece, there were satisfactory Cronbach’s α for all dimensions of TSI [30]. The Irritation Scale was validated in secondary education teachers, with a Cronbach’s overall α of 0.88 [31]. This is an instrument with eight items assessed using Likert-type responses ranging from 1 (strongly disagree) to 7 (strongly agree). In Germany, the scale has been used with several different samples from firefighters to psychologists, public employees and insurance companies. The studies carried out with these different populations offer very positive data regarding the reliability of the scale (for example, Cronbach’s α of 0.86 for industrial sector groups or Cronbach’s α of 0.91 for psychologists) [31]. An advantage of the present study is that all types of vocations from the Public Administration have been included, from the service sector to administrative positions, technical positions and manager positions. All the factors obtained a Cronbach’s α ≥ 0.700, and in all items, there was a correlation with their factor ≥0.300.

Other scales have focused on the response to chronic stress, such as the Marlach Burnout Inventory [32,33], designed to assess the frequency and intensity of perceived burnout among caregivers. It has been validated in Spanish teachers and university students [33]. Despite the wide international use of the Maslach Burnout Inventory, its psychometric properties have been questioned and alternative models have been suggested. For example, one study examined the psychometric properties and applicability of a Spanish version of the Maslach Burnout Inventory-Human Services Survey (MBI-HSS), in a sample of 947 social workers [34]. Other validated instruments for burnout are the occupational burnout scale in Mexico and the questionnaire for the evaluation of burnout syndrome at work (CESQT) [35,36]. Other instruments have focused on the stress generated by the job rotation in nursing professionals, such as the Nursing Job Rotation Stress Scale (NJRSS). Although it appears to be a reliable and valid instrument for evaluating job rotation stress, the sensitivity has not yet been adequately determined and it has only been validated in nurses [37].

Regarding, the use of ICTs at home and working at home, it should be noted that the use of computers and laptops outside of work hours has been related to work and family conflicts [38]. The European Agency for Safety and Health at Work identified technostress and risks associated with the increasing use of ICTs as the main new and emerging risk [39]. The use of work-related technologies is increasing, leading to a certain technological overload in all social areas [40]. The possibility of having contact with the employee at any time has a negative impact, creating the pressure of always having to be available and permitting work intrusions in the spaces and times that are normally reserved for private and family life. Indeed, workers face problems in separating work from private life [8]. A systematic review carried out on the effects of technostress revealed that the consequences of ICT use at home include feelings of tension, anxiety, exhaustion and decreased work satisfaction [41].

On the other hand, highly motivated employees tend to experience work-based conflicts since they often continue thinking about work even when they are at home. Labour autonomy, generally considered to be a positive resource, may in fact harm the employee. The urge to undertake excessive work may lead to an increased assignment of personal resources in the labor process, increasing household conflicts [42]. For these reasons, we not surprised to find that workers who think about work before going or had concerns about work at home, reported higher stress levels. The differences found in the association between satisfaction with the balance between work, personal life and work hours according to sociodemographic characteristics and the welfare system showed that there are inequalities in the work conditions within the different EU countries [43].

Finally, we observed correlations between the SSI-SM score and having a large dinner. This is in accordance with previous studies. For example, Suzuki et al. reported that those workers taking dinner after 21:00 felt stressed and tended to eat in excess [14]. Work-related stress may negatively impact the selection of foods thus contributing to poor health of the working population [44]. However, this is not necessarily related to BMI and, in fact, we did not find a statistically significant relationship between BMI and the overall SSI-SM score, or with any of the four factors, as it was in another study conducted on healthcare workers [45].

In addition, and this is a novelty of our study, we also did not find a significant relationship between high levels of stress according to the adapted SSI-SM and age, sex, civil status or the work position. According to the Spanish National Institute of Statistics, there are significant differences in stress according to sex, being higher in females than in males in all age ranges, except between 35–44 years [46]. Regarding the relationship between sex and work stress, stressors are more pronounced in females and have an unequal influence on males and females [47].

The American Psychological Association states that although females and males report similar average stress levels, females are more likely than males to report that their stress levels are increasing. They are also much more likely than males to report physical and emotional symptoms of stress. When comparing females to each other, there are also differences in the way married and single females experience stress, with married females reporting higher levels of stress than single females [48]. Thus, role overload could be an important factor to consider as a source of stress in females [49].

According to Cifre et al., work participation of both sexes continues to be different, the proportion of females who carry out precarious jobs is higher, as well as those who carry out part-time jobs [50]. Trying to balance work, family and housework may imply additional stress for women leading to physical and mental health problems. For this reason, the relationships between psychosocial exposures and work and life stress differ between men and women [51]. In our study, we only found a significantly higher level of stress in females regarding factor 3 “*Somatization*”, and a tendency towards the significance of factor 1 “*Self-concept*”. However, the overall score of the SSI-SM showed a non-significant higher perceived stress in females, which is probably related to the limited sample size of the study.

### 4.2. Limitations

There are some limitations in relation to this study. First, some of the study data is derived from self-reporting questionnaires. Although this is interesting since it reflects the real sense of every worker, it also may imply limitations inherent to data not obtained by an objective method. In addition, we did not assess the convergent validity or the stability (test-retest) of this adapted version of the SSI-SM. Moreover, only public employees from Murcia were considered in this study, and most were Caucasian. Thus, it is necessary to validate this scale with workers from other regions in Spain and other countries, due to the different employment contexts existing in other parts of the world, both Western and Eastern.

## 5. Conclusions

This study shows no significant association between high levels of stress according to the adapted SSI-SM and age, sex, BMI, civil status or the work position, in this cohort of public employees. However, we have validated an adapted version of the SSI-SM in a broad sample of public employees holding different work positions and shifts. After excluding two items, the Cronbach’s α of the adapted SSI-SM questionnaire was appropriate (0.909) and the factorial analysis demonstrated good homogeneity of the resulting factors (all with Cronbach’s α values > 0.700).

In Public Administration, different types of professionals from different sectors and different work shifts, carry out their work. Most current stress instruments focus on a particular profession. Thus, simple and friendly-use tools for the measurement of stress in public employees are necessary, such as the one we presented here. This tool may be of use by the Labour Risk Prevention Services for the early detection of work-related stress, thereby preventing some of its harmful consequences. Once stress has been detected, this information must be supplemented with other instruments. Nevertheless, it should be taken into account that there is a fine line between work and family, that workers tend to translate work and concerns to home, use computers before going to bed, not sleep correctly and altogether, do not allow the worker to “disconnect” from work, thus increasing stress.

## Figures and Tables

**Table 1 ijerph-17-08894-t001:** Factorial analysis of the questionnaire: rotated component matrix.

Kaiser-Meyer-Olkin Test Bartlett’s Test of Sphericity	0.930 < 0.001			
Items	Factor 1 Self-Concept	Factor 2 Sociability	Factor 3 Somatization	Factor 4 Uncertainty
I feel irritated	0.620			
I eat more or less than usual	0.511	0.412		
I leave things for another day				0.571
I am afraid	0.532			0.405
I am worried	0.731			
I am anxious	0.651			
I am defensive with others	0.589	0.440		
I am overwhelmed	0.644			
I have cold sweats			0.645	
My whole body itches			0.700	
I feel unable to take on the work				0.655
I feel indecisive				0.627
I neglect my social relationships		0.544		
I feel like I don’t know what to do		0.422		0.571
I have a negative attitude towards others		0.727		
I have heart palpitations			0.681	
I am angry	0.553	0.565		
I discredit others		0.556		
I have stomach aches			0.591	
I have difficulties relating to others		0.732		
**Self-values** **Variance**	7.660 41.29%	1.403 7.71%	1.183 6.21%	1.164 6.12%

**Table 2 ijerph-17-08894-t002:** Association between the stress questionnaire and personal and work-related factors of public employees.

Variables Investigated	Factor 1 Self-Concept	Factor 2 Sociability	Factor 3 Somatization	Factor 4 Uncertainty	Overall SSI-SM Score
Sex Males (*n* = 239) Females (*n* = 229)	15.0 ± 4.5 16.1 ± 5.3 *p* = 0.080	13.1 ± 4.1 13.4 ± 4.2 *p* = 0.702	5.0 ± 1.6 5.6 ± 2.1 *p* = 0.001	8.1 ± 2.3 8.2 ± 2.6 *p* = 0.241	33.1 ± 8.7 34.7 ± 9.8 *p* = 0.209
Civil status Single (*n* = 76) Married/living with partner (*n* = 355) Widow/er (*n* = 3) Divorced/Separated (*n* = 34)	16.1 ± 5.3 15.6 ± 5.0 16 ± 6.0 14 ± 3.7 *p* = 0.237	13.6 ± 4.4 13.3 ± 4.1 15 ± 4.5 12 ± 3.0 *p* = 0.219	5.3 ± 1.8 5.3 ± 1.9 6.6 ± 4.6 4.7 ± 1.0 *p* = 0.211	8.4 ± 2.6 8.1 ± 2.5 9.3 ± 3.2 7.6 ± 1.8.0 *p* = 0.328	34.7 ± 9.6 34.0 ± 9.4 37.3 ± 15.3 30.9 ± 6.2 *p* = 0.203
Age	R = − 0.009 *p* = 0.853	R = 0.200 *p* = 0.665	R = 0.032 *p* = 0.487	R = − 0.005 *p* = 0.910	R = 0.005 *p* = 0.915
BMI	R = 0.060 *p* = 0.197	R = 0.023 *p* = 0.615	R = 0.004 *p* = 0.937	R = 0.024 *p* = 0.600	R = 0.045 *p* = 0.334
Work position Managers (*n* = 59) University technicians (*n* = 279) Administrators (*n* = 96) Manual positions (*n* = 35)	15.3 ± 4.7 15.9 ± 5.2 15.3 ± 4.9 14.4 ± 3.4 *p* = 0.321	13.2 ± 4.3 13.4 ± 4.2 13.3 ± 4.2 12.3 ± 3.3 *p* = 0.525	4.9 ± 1.3 5.5 ± 2.1 5.0 ± 1.6 5.0 ± 1.4 *p* = 0.009	7.8 ± 2.5 8.3 ± 2.6 7.9 ± 2.2 7.8 ± 2.1 *p* = 0.235	33.0 ± 8.8 34.7 ± 9.8 32.9 ± 8.6 31.8 ± 7.2 *p* = 0.152
I stay in bed, even when I cannot fall asleep	R = 0.014 *p* = 0.770	R = 0.001 *p* = 0.979	R = 0.057 *p* = 0.218	R = − 0.10 *p* = 0.837	R = − 0.002 *p* = 0.965
I think about work at home, or have concerns regarding work before going to bed	R = 0.360 *p* < 0.001	R = 0.331 *p* < 0.001	R = 0.234 *p* < 0.001	R = 0.318 *p* < 0.001	R = 0.374 *p* < 0.001
I do some home-based telework	R = 0.250 *p* < 0.001	R = 0.191 *p* < 0.001	R = 0.148 *p* < 0.001	R = 0.292 *p* < 0.001	R = 0.264 *p* < 0.001
I use to take a copious or oversized dinner	R = 0.178 *p* < 0.001	R = 0.162 *p* < 0.001	R = 0.079 *p* = 0.089	R = 0.229 *p* < 0.001	R = 0.200 *p* < 0.001
I use ICTs resources at home	R = 0.125 *p* = 0.007	R = 0.69 *p* = 0.069	R = − 0.057 *p* = 0.236	R = 0.161 *p* < 0.001	R = 0.100 *p* = 0.031

R: Pearson’s correlation coefficient *p*: statistical significance.

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
