# Peer review of "Reliability and Validity of a Stress Scale in Public Employees from Murcia (Spain)"

_ijerph, 2020, doi:10.3390/ijerph17238894_

Round 1
Reviewer 1 Report
Dear Authors,
I appreciate having the opportunity to review the manuscript entitled “Reliability and validity of a stress scale in Spanish public employees” (ijerph-954924).
Although the authors have made considerable efforts to develop this interesting paper, however, I believe that the current version of manuscript should be improved through significant revision and re-writing. I want to provide some suggestions for the improvement of this paper as follows.
[1] Introduction
- I think that the overall structure and writing of introduction part are not clear and well-aligned because it is not easy to catch what the overall structure and strategies of this paper are. Please clearly describe those things. As you already knew, the introduction section is one of the most important parts to not only draw attentions of readers but also provide guidelines for them to facilitate a clear understanding of the paper.
[2] Theories and hypotheses
- It is difficult for me to be sure that the research has an enough level of theoretical value and contribution. I think that this is the critical flaw of this paper. Please provide the part in an elaborated way.
- Although this paper dealt with interesting phenomena, it did not provide adequate theoretical background and support for the development of its hypotheses. This is the critical limitation of this paper. Please clearly explain what its hypotheses are.
[3] Analytical Strategy
- I think that the authors should provide the result of confirmatory factor analysis (CFA) after conducting the factor analysis.
I wish these comments may help you to improve your paper. Good luck.
Author Response
Dear Authors,
I appreciate having the opportunity to review the manuscript entitled “Reliability and validity of a stress scale in Spanish public employees” (ijerph-954924).
Although the authors have made considerable efforts to develop this interesting paper, however, I believe that the current version of manuscript should be improved through significant revision and re-writing. I want to provide some suggestions for the improvement of this paper as follows.
>>> Thanks your comments and suggestions. They helped us to improve our manuscript.
[1] Introduction
I think that the overall structure and writing of introduction part are not clear and well-aligned because it is not easy to catch what the overall structure and strategies of this paper are. Please clearly describe those things. As you already knew, the introduction section is one of the most important parts to not only draw attentions of readers but also provide guidelines for them to facilitate a clear understanding of the paper.
>>> Thanks for this suggestion. The introduction has been modified and we think it is now clearer, better focused and easier to read.
[2] Theories and hypotheses
It is difficult for me to be sure that the research has an enough level of theoretical value and contribution. I think that this is the critical flaw of this paper. Please provide the part in an elaborated way.
Although this paper dealt with interesting phenomena, it did not provide adequate theoretical background and support for the development of its hypotheses. This is the critical limitation of this paper. Please clearly explain what its hypotheses are.
>>> Thanks for both comments. We certainly agree with you and therefore we have included information in the Introduction section addressing this issue. We hope this revised version of the manuscript would clarify our hypothesis underlying the development of the study.
[3] Analytical Strategy
I think that the authors should provide the result of confirmatory factor analysis (CFA) after conducting the factor analysis.
>>> We appreciate this suggestion. However, as this was a pilot study and given the limited time and the complicated situation we are currently living in our region due to the COVID-19 pandemic, we were not able to provide such analysis this time. We are really sorry about this issue, and we tried to improve, clarify, and enhance the paper as much as we can, and we hope this would be adequate for the reviewer.
I wish these comments may help you to improve your paper. Good luck.
>>> For sure they did. Thanks!
Reviewer 2 Report
Dear authors,
I would like to congratulate you for your study, which I considered quite interesting. Stress in work environments is a not a novel topic, but it is becoming more and more relevant.
Even though I think the paper presents potential for publication, some issues should be addressed:
1) The Abstract is not structured, so, you should be careful when writing it. For instance, you should not state "To determine the internal consistency", but "The aim of this study was to (...)".
2) The Introduction section should present additional information regarding the tools which already exist to assess stress. You must present a gap, i.e., you must argue for the need of a new assessment tool having into account the analysis of the tools which already exist.
3) When reading the items of the SSI-SM, I did not find any item that evaluates aspects directly related to the work environment. Considering that, you should state why this tool is needed, what does it add in comparison with the tools that already exist.
4) At the Materials and Method section, it is not clear how did you collect data. Was it online? Face to face? What was the sampling technique? You should also justify your option regarding the sampling technique.
5) Why 468 public employees? Did you previously calculate the sample you needed for carrying out this study? Or the sample was not intended to be representative?
6) Also in the Materials and Method section, you should describe more clearly the assessment tool. Has it been validated in many countries? What were its psychometric properties?
7) Why did you decide not to assess, for instance, the convergent validity or the stability (test-retest) of the tool?
8) Your methodological options should be grounded on the literature. Thus, you should cite authors which consubstantiates your methodological options.
9) You should also discuss the findings that did not present statistically significant differences, not only the ones that present statistically significant differences.
10) Your discussion about the psychometric properties of the assessment tool is a bit poor. You should compare and contrast its psychometric properties with the ones of other stress assessment tools.
11) I am not sure only collecting data in Spain is a limitation, considering the aim of your study was to validate the assessment tool. Perhaps there are other limitations in your study which are not properly stated (related to the sample / sampling technique, for instance).
12) Your conclusions do not answer the two objectives of your research. Moreover, at the Conclusion section you should present recommendations for further research, as well as the implications / contributes of your study for nursing (clinical practice, education and/or research).
Author Response
Dear authors,
I would like to congratulate you for your study, which I considered quite interesting. Stress in work environments is a not a novel topic, but it is becoming more and more relevant.
Even though I think the paper presents potential for publication, some issues should be addressed:
>>> Thank you so much for you overall positive opinion about our study.
1) The Abstract is not structured, so, you should be careful when writing it. For instance, you should not state "To determine the internal consistency", but "The aim of this study was to (...)".
>>> Thank you for this suggestion. We agree with you and therefore, we have corrected the abstract and structured each section adding introductory phrases such as "the aim of this study", "results reveal" and “these data show”.
2) The Introduction section should present additional information regarding the tools which already exist to assess stress. You must present a gap, i.e., you must argue for the need of a new assessment tool having into account the analysis of the tools which already exist.
>>> Thanks for this suggestion. The introduction section has been modified and more information regarding this issue has been included.
3) When reading the items of the SSI-SM, I did not find any item that evaluates aspects directly related to the work environment. Considering that, you should state why this tool is needed, what does it add in comparison with the tools that already exist.
>>> This is an interesting comment. The reviewer is correct; there is no work environment item in the SSI-SM. There are three perspectives of stress, as a stimulus, as a response, and as an event-reaction relationship. In the present study, we aimed to measure stress as a response, i.e. a psychological, physiological or behavioral response that is based on the theory of Selye. Under this perspective, stress generates somatic, emotional and behavioral symptoms and this causes negative effects in the organization. With the SSI-SM questionnaire, we measured the symptoms of the individual, not the work environment. In other words, we were focused on the worker and not on the work.
In our opinion, this is the first step of a two steps process. The first one is to identify the manifestations of stress in workers, whereas the second one is to focus on the work/organization environment and potential stressors within the work.
4) At the Materials and Method section, it is not clear how did you collect data. Was it online? Face to face? What was the sampling technique? You should also justify your option regarding the sampling technique.
>>> We apologize for this issue. We have clarified the Methods section overall, including supporting information and re-structured the section.
5) Why 468 public employees? Did you previously calculate the sample you needed for carrying out this study? Or the sample was not intended to be representative?
>>> We are sorry for not including this information in our first version. This has now been added in the first paragraph under the subsection statistical analysis.
6) Also in the Materials and Method section, you should describe more clearly the assessment tool. Has it been validated in many countries? What were its psychometric properties?
>>> Thank you and apologies if there was limited information about the SSI-SM in the previous version of our manuscript. We have added some details about this in the Materials and Method section.
7) Why did you decide not to assess, for instance, the convergent validity or the stability (test-retest) of the tool?
>>> This is an interesting suggestion. When we performed the validation of the scale, we did not consider it was necessary and therefore we did not proceed with such analysis. For this reason, we have included this as a limitation since it would be very interesting for future studies. Thanks for your appreciation.
8) Your methodological options should be grounded on the literature. Thus, you should cite authors which consubstantiates your methodological options.
>>> Thank you. We have updated the Methods section with appropriate references.
9) You should also discuss the findings that did not present statistically significant differences, not only the ones that present statistically significant differences.
>>> Thank you for this interesting suggestion. We have added some text and references in the Discussion section in relation to this issue.
10) Your discussion about the psychometric properties of the assessment tool is a bit poor. You should compare and contrast its psychometric properties with the ones of other stress assessment tools.
>>> We agree with the reviewer. The discussion section has been expanded with information and evidence regarding this issue.
11) I am not sure only collecting data in Spain is a limitation, considering the aim of your study was to validate the assessment tool. Perhaps there are other limitations in your study which are not properly stated (related to the sample / sampling technique, for instance).
>>> We apologize for this. We have improved this subsection and included some other limitations required to be acknowledged.
12) Your conclusions do not answer the two objectives of your research. Moreover, at the Conclusion section you should present recommendations for further research, as well as the implications / contributes of your study for nursing (clinical practice, education and/or research).
>>> We are sorry for this issue. In this revised version we have modified the conclusion and we think it is now clearer and includes the information requested by the reviewer.
Reviewer 3 Report
Introduction:
- Literature review needs to be updated to include recent empirical research done in this area. There are some citations from 2019 and 2020 but mostly based on internet articles, not empirical literature. Lot of quotes studies in review seem outdated.
- Some concepts need to be defined more in the introduction section, such as "mood swings" (line 43), "labour dissatisfaction" (line 56), "personal and labor factors" (line 68).
- Line 58 needs to be reworded, as "decrease coping strategies" is unclear. It might be helpful to differentiate the use of adaptive vs maladapative coping strategies.
Methods:
- The term personal and labour factors sound too broad for the purpose of the study, and its operational definition needs to be provided for the purpose of this study.
- Did the workers receive any compensation for their participation in the study?
- Please consider revising the terminology for factors being studied that contribute to work related stress, namely "stay in bed when unable to fall asleep" and "eat large dinner". Also it would be helpful to provide reasoning for how and why these factors were chosen to be studied. Also, were the factors defined for the participants e.g. how long would they need to stay in bed, or how much is a "large" portion of dinner?
Results and discussion:
- Discussion section needs to be presented in a more organized manner.
- In the discussion section, please provide possible explanations for the results obtained in the study.
Conclusion:
Please say more about the implications of the study for public workers and provide specific recommendations regarding how workers prone to stress could be helped to prevent burnout.
Author Response
Introduction:
Literature review needs to be updated to include recent empirical research done in this area. There are some citations from 2019 and 2020 but mostly based on internet articles, not empirical literature. Lot of quotes studies in review seem outdated.
>>> We apologize for this. We have included some recent references. Please, note that we have updated the references since most but were actually empirical literature and not internet articles, but were presented in other format.
Some concepts need to be defined more in the introduction section, such as "mood swings" (line 43), "labour dissatisfaction" (line 56), "personal and labor factors" (line 68).
>>> The concepts negative mood and job satisfaction have been clarified in the Introduction, we apologize if they could be misleading. Personal and work-related factors investigated have also been clarified and are detailed in the Methods section.
Line 58 needs to be reworded, as "decrease coping strategies" is unclear. It might be helpful to differentiate the use of adaptive vs maladapative coping strategies.
>>> Updated. Thank you.
Methods:
The term personal and labour factors sound too broad for the purpose of the study, and its operational definition needs to be provided for the purpose of this study.
>>> We apologize if this was a bit unclear in the previous version. We have now clarified this information in the Methods section.
Did the workers receive any compensation for their participation in the study?
>>> No. Participation in the study was completely voluntary and there was no compensation for participation.
Please consider revising the terminology for factors being studied that contribute to work related stress, namely "stay in bed when unable to fall asleep" and "eat large dinner". Also it would be helpful to provide reasoning for how and why these factors were chosen to be studied. Also, were the factors defined for the participants e.g. how long would they need to stay in bed, or how much is a "large" portion of dinner?
>>> We have clarified such terms in the Methods section of the manuscript. We apologize if they were not clear in the previous version.
Also, we included these items because previous studies have shown and association with stress. There is a significant association between sleep quality and stress. A high level of stress is an important predictor of poor sleep quality. Sleep and stress interact bidirectionally, sharing multiple pathways that affect the central nervous system and metabolism. Importantly, stress hormone levels are positively correlated with decreased sleep duration. On the other hand, copious or overeating dinners are related to responses to stress. Suzuki et al. found that those who took dinner after 21:00 and felt stressed, tended to overeat during dinner. In this study, factors related to overeating at dinner included response to psychological stress and dinner time.
Results and discussion:
Discussion section needs to be presented in a more organized manner.
In the discussion section, please provide possible explanations for the results obtained in the study.
>>> We have corrected and re-structured the discussion. We hope this would be better and easy to read. Thank you.
Conclusion:
Please say more about the implications of the study for public workers and provide specific recommendations regarding how workers prone to stress could be helped to prevent burnout.
>>> In this revised version we have modified the conclusion and we think it is now clearer and includes the information requested by the reviewer.
Reviewer 4 Report
Dear Authors.
When reading your work, some doubts have come up that I would like you to clarify in the article:
Why was the student questionnaire administered if the study was with employees? There is no other questionnaire to specifically measure stress among Spaniards. Please review the literature
There is a lack of information on how the grantees were managed and how the sample was obtained.
How long did it take them to fill out the questionnaires?
Why did it take more than 2 years to use these questionnaires with 468 employees? Is this study part of another study?
What is the code of ethics committee?
In the limitations part I would add that this is a sample that represents workers from Murcia, but not all of Spain. Therefore it would be interesting not only to do it in other countries but in all of Spain as well.
Thank you
Translated with www.DeepL.com/Translator (free version)
Author Response
Dear Authors.
When reading your work, some doubts have come up that I would like you to clarify in the article:
Why was the student questionnaire administered if the study was with employees? There is no other questionnaire to specifically measure stress among Spaniards. Please review the literature
>>> Work-related stress depends on both the objective psychosocial conditions and the worker's interpretation of them. The simultaneous use of various evaluation methods and techniques may be convenient. Work stress can be evaluated using methods that analyze the different work conditions as psychosocial risk factors, to identify work stressors and to intervene on them, and in a complementary way to measure their impact on the worker's health with the general health questionnaire. Instruments for measuring work stress in Spanish, focus on stress as a perceived stimulus but they do not take into account the response to stress. A different meaning of the term stress is to consider it as the stimulus, the casual agent of that response, ie. the stressor.
As we have now included in the Discussion section, there are other instruments/tools to measure stress, and some have been validated in a Spanish population. However, to the best of our knowledge, this is the first one validated in public employees. Our aim was to validate a simple tool to measure stress and we considered that the SSI-SM, after a short adaptation would be useful.
There is a lack of information on how the grantees were managed and how the sample was obtained.
>>> We have expanded this information in the Methods section.
How long did it take them to fill out the questionnaires?
>>> There was no a specific time to fulfill the questionnaires. All patients took as much time as they required. However, these were completed during the medical examinations by the Risk Prevention Service, which lasted 30 minutes approximately.
Why did it take more than 2 years to use these questionnaires with 468 employees? Is this study part of another study?
>>> There is no a particular reason for this. All public workers were included when attended routine medical examinations by the Risk Prevention Service However, the Risk Prevention Service performs medical examinations only during two short periods every year. Thus, 2 years were necessary to include an appropriate sample size.
What is the code of ethics committee?
>>> We have added this information in the Methods section.
In the limitations part I would add that this is a sample that represents workers from Murcia, but not all of Spain. Therefore it would be interesting not only to do it in other countries but in all of Spain as well.
>>> We have expanded the limitations and acknowledge this. Thank you.
Thank you
Round 2
Reviewer 2 Report
Dear authors,
Thank you for having accepted some of my previous recommendations. I think that, overall, the paper is better than in its former version. However, some issues still need to be addressed:
1) At the "Introduction" section you explained the existence of a gap which justifies the need of your study, and that is OK. However, your approach to the assessment tools that already exist is poorly comprehensive. You should clearly present what assessment tools already exist, what are the differences between them, why none of them is good enough to assess the stress of public employees,...
2) You assumed, in your answer, there are no work environment items in the SSI-SM. You also stated that you aimed to measure stress as a response. My question is: are there no other assessment tools, in Spain, which allow to assess stress as a response? Why do you consider assessment tools such as, for instance, the Perceived Stress Scale, cannot do that? Please, make it clear in the paper, and not only as an answer to the reviewer, because that is a relevant information.
3) You stated the sample size was calculated based on an estimation of 42% of perceived stress. How did you reach/find that value?
4) In the title you define your study as a pilot study. At the Abstract and at the Materials and Methods sections you define it as a descriptive and cross-sectional study. That is a non-sense. In fact, I cannot look at your paper as a pilot study, and I do not understand why you classified it that way. Moreover, indeed, your study is not only descriptive, but also correlational.
5) You added to the Discussion section some ideas about the absence of statistically significant differences between sex and stress levels in your study, which is not in line with previous studies. However, you should interpret that finding. Why do you think / How do you explain that in your study there were no statistically significant differences between stress level in male and female workers?
6) Why did you consider having collected data mostly in Caucasian employees is a limitation? Is there any literature pointing out to differences in stress level between Caucasian vs. non-Caucasian?
7) At the Conclusion section you only present an answer to one of the objectives of your study. Thus, in order to also answer the other objective, you should present the psychometric properties of the scale according to the findings of your study.
Author Response
Reviewer 2
Dear Authors,
Thank you for having accepted some of my previous recommendations. I think that, overall, the paper is better than in its former version. However, some issues still need to be addressed:
>>> Thank you.
1) At the "Introduction" section you explained the existence of a gap which justifies the need of your study, and that is OK. However, your approach to the assessment tools that already exist is poorly comprehensive. You should clearly present what assessment tools already exist, what are the differences between them, why none of them is good enough to assess the stress of public employees,...
>>> We agree with the reviewer and we thank for this interesting suggestion. To date, there are no questionnaires, aimed at all types of workers, focusing on stress as a response. However, we have included more information in relation to this issue in the Introduction section. The Perceived Stress Scale is the most known tool in Spain, but different studies have shown disparate results regarding psychometric properties.
2) You assumed, in your answer, there are no work environment items in the SSI-SM. You also stated that you aimed to measure stress as a response. My question is: are there no other assessment tools, in Spain, which allow to assess stress as a response? Why do you consider assessment tools such as, for instance, the Perceived Stress Scale, cannot do that? Please, make it clear in the paper, and not only as an answer to the reviewer, because that is a relevant information.
>>> See response to the first comment. Thank you.
3) You stated the sample size was calculated based on an estimation of 42% of perceived stress. How did you reach/find that value?
>>> We apologize for not including the appropriate references in the previous version, and we are sorry because we have identified a mistake in the figure, the correct one is 45%, not 42%. This value was reached by taking into account the prevalence of stress in a study based on Latin population, in a study in Spanish public workers from a local administration, and using the mean stress in Spain for all workers according to the Spanish National Institute of Statistics. These three references have been included in the manuscript.
4) In the title you define your study as a pilot study. At the Abstract and at the Materials and Methods sections you define it as a descriptive and cross-sectional study. That is a non-sense. In fact, I cannot look at your paper as a pilot study, and I do not understand why you classified it that way. Moreover, indeed, your study is not only descriptive, but also correlational.
>>> We apologize for this and we agree with you actually. We have deleted the statement of ‘pilot study’ in the title. Thank you.
5) You added to the Discussion section some ideas about the absence of statistically significant differences between sex and stress levels in your study, which is not in line with previous studies. However, you should interpret that finding. Why do you think / How do you explain that in your study there were no statistically significant differences between stress level in male and female workers?
>>> Thank you for this comment. In our opinion, there is no particular reason for such observation, however, we think it could be related to the small sample size since the overall score of the SSI-SM showed a score >1.5 points over in females than in males.
6) Why did you consider having collected data mostly in Caucasian employees is a limitation? Is there any literature pointing out to differences in stress level between Caucasian vs. non-Caucasian?
>>> Thanks for your comment. Yes, we should recognize that including a population mainly Caucasian may be a limitation for extrapolating our results to others region of the world where the proportion of other ethnics and razes are higher. For example, there are studies suggesting that African Americans may be particularly vulnerable to the burden of chronic stress compared to Caucasians (see Troxel et al. Chronic stress burden, discrimination, and subclinical carotid artery disease in African American and Caucasian women. Health Psychology 2003; 22 (3): 300-309). Thus, we consider this statement should be included.
7) At the Conclusion section you only present an answer to one of the objectives of your study. Thus, in order to also answer the other objective, you should present the psychometric properties of the scale according to the findings of your study.
>>> We are sorry for this and agree with you. We have added some of this information in the conclusion section. Thanks